# Asymmetric rotaxanes as dual-modality supramolecular imaging agents for targeting cancer biomarkers

Faustine d'Orchymont[1] & Jason P. Holland [1]✉

Dual-modality imaging agents featuring both a radioactive complex for positron emission tomography (PET) and a fluorophore for optical fluorescence imaging (OFI) are crucial tools for reinforcing clinical diagnosis and intraoperative surgeries. We report the synthesis and characterisation of bimodal mechanically interlocked rotaxane-based imaging agents, constructed via the cucurbit[6]uril CB[6]-mediated alkyne-azide 'click' reaction. Two synthetic routes involving four- or six-component reactions are developed to access asymmetric rotaxanes. Furthermore, by using this rapid and versatile approach, a peptide-based rotaxane targeted toward the clinical prostate cancer biomarker, prostate-specific membrane antigen (PSMA), and bearing a $^{68}$Ga-radiometal ion complex for positron emission tomography and fluorescein as an optically active imaging agent, was synthesised. The chemical and radiochemical stability, and the cellular uptake profile of the radiolabelled and fluorescent rotaxane was evaluated in vitro where the experimental data demonstrate the viability of using an asymmetric rotaxane platform to produce dual-modality imaging agents that specifically target prostate cancer cells.

[1] University of Zurich, Department of Chemistry, Winterthurerstrasse 190, CH-8057 Zurich, Switzerland. ✉email: jason.holland@chem.uzh.ch

There is a high degree of interest in developing dual-modality agents that can be used to detect biomarkers of disease in combination with more than one imaging technique. The concept of dual-modality positron emission tomography (PET)/optical fluorescence imaging (OFI) for in vivo applications hinges on the development of compounds that contain a radionuclide, a fluorophore, and a biological targeting vector within one construct[1,2]. When the two functionalities are combined within a single molecular entity, potential issues regarding differences in the pharmacokinetics of two separate imaging agents can be avoided. OFI has shown great potential in diagnostic applications, where it is used frequently in both living cells and in the analysis of tissue samples ex vivo[3–11]. The ability to perform intraoperative monitoring is an important feature of OFI[12] which can be applied in real-time imaging-guided surgery to identify tumour margins and facilitate tumour resection[4,13]. Small-molecule fluorophores are often built around five molecular scaffold classes, which include coumarin, BODIPY, fluorescein, rhodamine, and cyanine dyes, all of which have been used for fluorescence-based imaging[14–16]. Despite its high sensitivity, fluorescence imaging has several limitations in vivo, mainly associated with limited quantification as a result of the poor (and wavelength-dependent) penetration depth of light through tissue, which is absorbed and scattered in the body[17]. The use of OFI in combination with PET imaging in dual-modality agents has the potential to reinforce intraoperative detection of cancerous tissues by improving the localisation and facilitating surgical removal of lesions through enhanced delineation of tumour margins[18]. In addition, the sensitivity of OFI and PET probes is comparable. Therefore, the use of a 1:1 stoichiometric ratio of each imaging component is feasible, and only a trace amount of the agent is required for applications in vivo[1].

Dual-labelled antibody- or peptide-based probes that specifically target prostate cancer cells are well-described in the literature[17–22]. Compelling examples are the dual-modality probes based on PSMA-11 evaluated by Baranski et al. for imaging in vivo[23]. PSMA-11 is a clinically validated [68Ga]-labelled radiotracer based on the HBED-CC chelate and a small-molecule Lys-urea-Glu peptidic-inhibitor that binds to prostate-specific membrane antigen (PSMA)[24]. Four different fluorophores conjugated to the second (free) carboxylate arm of the HBED-CC-Ahx-Lys-urea-Glu ligand were evaluated. The [68Ga]-radiolabelled dye conjugates tested displayed affinity for PSMA in the nanomolar range and high PSMA-specific uptake in tumours, comparable to the parent [[68Ga]Ga-PSMA-11 complex in LNCaP cell-binding assays and in small-animal PET imaging[25]. Additionally, specific cell binding was visualised by confocal microscopy, and successful fluorescence-guided prostatectomies were also performed 1 h post-injection of the IRDye800CW dye conjugate on healthy pigs[23]. These data highlight the potential of dual-modality PET/OFI ligands for intraoperative detection of prostate cancer. Nevertheless, the design and synthesis of dual-modality imaging agents is non-trivial.

Supramolecular systems and mechanically interlocked molecules (MIMs) present interesting opportunities in the development of dual-modality imaging agents[26–30]. Specifically, the use of mechanical bonding in tracer design provides new opportunities for controlling the pharmacokinetics and metabolism of imaging agents in vivo[31–33]. In addition, the self-assembly mechanisms used in the rapid synthesis of supramolecular compounds like rotaxanes, molecular cages, or catenanes offer new routes for controlling the stoichiometry and molecular architecture of imaging probes. However, to date, only a few studies have explored the potential of using supramolecular compounds as imaging probes. Most of the reported work is centred on the use of supramolecular chemistry to develop single modality agents[32–36]. Recently, we developed viable supramolecular PET imaging agents based on a rotaxane scaffold synthesised via a cooperative capture strategy involving two macrocycles, β-cyclodextrin (β-CD) and cucurbit[6]uril (CB[6]). Our rotaxane-based radiotracers featured [89Zr]- or [68Ga]-radiolabelled metal-ion binding chelates for PET and were conjugated to monoclonal antibodies for cancer-specific localisation[26]. Here, we present the synthesis of supramolecular dual-modality agents by expanding on the rotaxane platform to access bimodal imaging agents for applications in PET/OFI. Two parallel synthetic approaches, which feature discrete control over the number of components involved in the cooperative capture

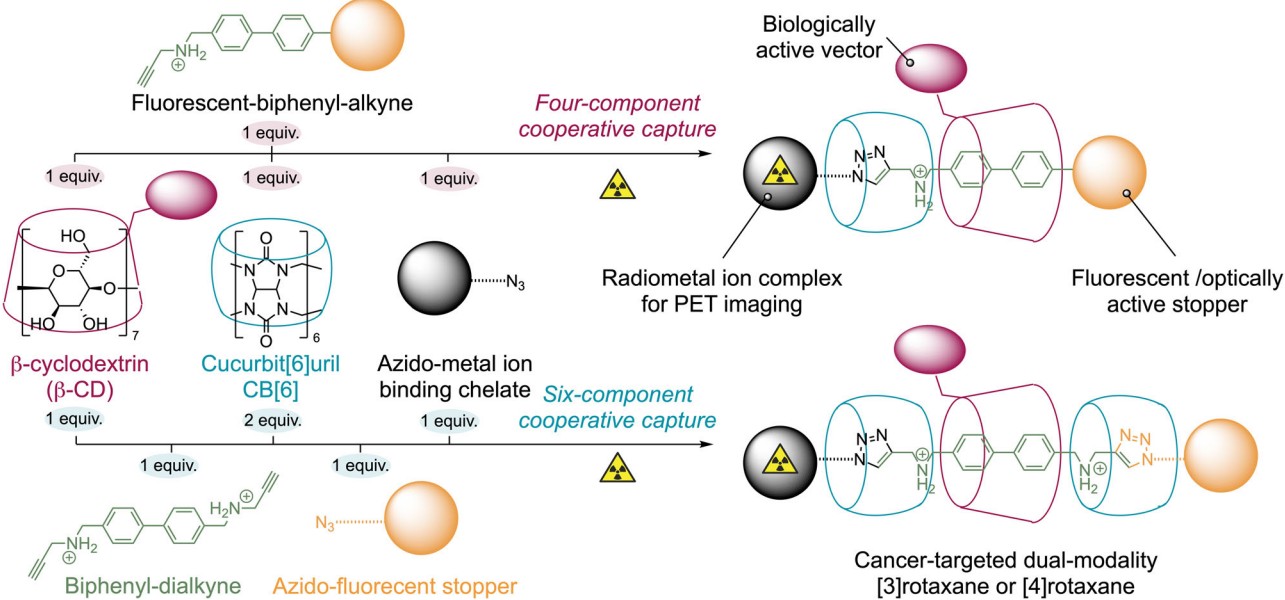

**Fig. 1 Synthetic scheme describing the two strategies for the synthesis of [68Ga]-labelled-bifunctionalised rotaxanes.** Note: the radionuclide is incorporated in a second step after the isolation of the semirotaxane. (Top) The minimalist four-component pathway involves precisely one stoichiometric equivalent of each reagent. (Bottom) A six-component variant involving a double CB[6]-accelerated triazole synthesis.

step (four or six components, Fig. 1), are described as leading to the formation of [3]rotaxanes or [4]rotaxanes. Our goal was to develop multimodality PET/OFI rotaxane conjugates that target prostate cancer by attaching a fluorescent probe and a [68]Ga-radiolabelled metal-ion binding chelate to the biologically active PSMA targeting vector Lys-urea-Glu (Fig. 1). The PSMA targeting vector was integrated on the β-CD macrocyclic host and mechanically bonded to the dual-modality rotaxane axle. One end of the axle was capped by the metal-ion binding chelate, which can be radiolabelled, and the other end was used to incorporate a fluorescein derivative that simultaneously acts as an optical probe and as a capping agent for the rotaxane.

## Results and discussion

The original cooperative capture strategy using the CB[6]-mediated click reaction was first reported in 1983 by Mock et al.[37–40]. The method was later developed by Stoddart and co-workers and has been used extensively in the synthesis of pseudorotaxanes, semirotaxanes, rotaxanes, and polyrotaxanes[26,41–44]. A crucial feature for the successful and rapid synthesis of molecularly interlocked molecules using the cooperative capture method is the ability of a guest molecule to form an inclusion complex with the β-CD derivative. Previously, we showed that asymmetric bifunctional [3]rotaxanes could be synthesised[26], but the efficiency of the reaction and the mixture of products delivered were not optimal. Low yields and the lack of product control were due to the very low binding constant for the inclusion complex between a fluorescein-PEG$_2$-biphenyl unit (the guest molecule) and β-CD (the host molecule, binding constant $K = 167 \pm 70$ M$^{-1}$ calculated from [1]H NMR titration data). In addition, in our original design, the extended PEG$_2$ linker provided the possibility for the β-CD to shuttle between different positions along the axle. This movement between binding sites removed the cooperativity between β-CD and CB[6] macrocycles, which reduced the rate of triazole formation via the CB[6]-mediated click reaction. Indeed, our reaction rates were consistent with the original observations of CB[6]-catalysed triazole formation by Mock et al.[38] which confirmed the lack of involvement of the β-CD macrocycle.

Following on from these preliminary results, it was evident that optimisation of the linker design was essential to restore the β-CD and CB[6] interactions and thereby enhance the rate of the [3]semirotaxane formation. A new fluorescein-derivatised biphenyl guest molecule (1) was synthesised from a propargyl amine-acetal substituted biphenyl intermediate and fluorescein over a 3-step synthetic procedure (see supporting information, Scheme S1 and Figs. S1–S13). Compound 1 was isolated with an overall yield of 20%. We opted for a relatively short ethylenediamine group as a linker between the fluorescein and the biphenyl unit. This ethylenediamine group is shorter than the PEG$_2$ linker previously installed, which should potentially increase the intermolecular interaction between the β-CD and CB[6] macrocycles and increase the rate of the CB[6]-accelerated click reaction. Prior to further experiments with compound 1, a salt exchange was performed with 1 M HCl, and after lyophilisation, the samples were redissolved in H$_2$O or D$_2$O.

Before synthesising the rotaxanes, we measured the binding affinity and stoichiometry for the molecular inclusion of 1 inside the cavity of β-CD by using two [1]H NMR titration methods. As depicted in Fig. 2a, shifts in the resonance frequencies of five protons were considered on compound 1: H$_a$ and H$_b$ on the biphenyl moiety and H$_c$, H$_d$ and H$_e$ on each side of the ethylenediamine linker (Fig. 2b). In addition, the chemical shifts observed for five distinct proton resonance peaks assigned to the β-CD macrocycle were measured (Supporting information Fig. S14). The proton resonance peaks of the ethylene linker could

not be considered in the numerical analysis because they overlap with several β-CD resonance peaks in the [1]H NMR spectrum. A plot showing the measured change in chemical shifts observed for the guest molecule is presented in Fig. 2c, and equivalent data for the β-CD resonance peaks are given in Supporting Information Fig. S15. By using the method of continuous variation, the Job's plots were plotted for the selected proton resonance peaks of the guest molecule 1 and of β-CD (Supporting Information Fig. S16). Analysis of the Job's plots showed a maximum at $r = 0.5$ for all protons, confirming the presence of the 1⊃β-CD molecular inclusion complex in solution. Notably, since all five protons selected on 1 showed a Job's plot maximum at 0.5, the data indicate that the position of the β-CD macrocycle was in rapid equilibrium between the biphenyl moiety and the ethylenediamine linker. Therefore, the data confirmed that the β-CD macrocycle could slide on the axle of the guest molecule (Fig. 2a). In addition, due to the asymmetry of the guest molecule 1, two equally probable conformations of the host–guest complex occur in which the β-CD orientation in the 1⊃β-CD inclusion complex can invert between either the primary or secondary face of the β-CD macrocycle facing the fluorescein unit (Fig. 2a). Based on our spectroscopic data, we can conclude that this sliding equilibrium and β-CD face inversion is fast relative to the NMR time scale.

All five protons selected on 1 were also considered for the determination of the binding constant of the 1⊃β-CD inclusion complex. The [1]H NMR spectra recorded when β-CD was gradually added to a fixed concentration of 1 (0.25 mM) are displayed in Fig. 2b. From the chemical shift variation ($\Delta\delta_H$) between a given proton resonance peak on the free guest molecule and the same proton when complexed to β-CD, analysis by using the Benesi–Hildebrand[45], Scott[46], and Scatchard[47] methods for a 1:1 equilibrium afforded linear plots with correlation coefficients ($R^2$) between 0.8456 and 0.9945 (Fig. 2d and Supporting Information Fig. S17). The binding constants extracted from the three models showed some minor disparities (Fig. 2d). Averaging across data obtained for the resonance shifts observed for the measured protons on the guest molecule (H$_a$–H$_e$) gave average binding affinities of $K = 759 \pm 183$ M$^{-1}$ (Benesi–Hildebrand), $K = 473 \pm 240$ M$^{-1}$ (Scott), $K = 407 \pm 131$ M$^{-1}$ (Scatchard) for the three different models. Averaging across the 3 models for each proton gave binding constant values of $K = 684 \pm 178$ M$^{-1}$ for H$_a$, $388 \pm 110$ M$^{-1}$ for H$_b$, $420 \pm 119$ M$^{-1}$ for H$_c$, $710 \pm 141$ M$^{-1}$ for H$_d$, and $443 \pm 114$ M$^{-1}$ for H$_e$. Although no clear trend could be extracted from these data, with all binding constants being in the same range (average binding constant from the three models $K = 557 \pm 329$ M$^{-1}$), the numbers were only slightly lower than the binding constant determined for the symmetrical biphenyl alkyne 2 (vide infra, Fig. 3a) using the same methods ($K = 736 \pm 85$ M$^{-1}$)[26]. However, we note that this binding constant for the 1⊃β-CD inclusion complex was 3.4-fold higher than the value obtained previously when using a fluorescein-biphenyl derivative featuring a longer PEG$_2$ linker ($K = 167 \pm 70$ M$^{-1}$)[26]. These data indicated that 1 is a potentially suitable reagent for use in the cooperative capture synthesis of rotaxanes.

Next, we evaluated the use of the fluorescein-alkyne guest molecule 1 in the cooperative capture synthesis of asymmetric rotaxanes via the CB[6]-β-CD-azide-alkyne cycloaddition reaction (Fig. 3a and Supporting Information Scheme S2). Compound 1 was mixed with β-CD, CB[6] and an azido derivative of the desferrioxamine B chelate (DFO-azido, 3) in a 1:1:1:1 ratio in H$_2$O and heated at 70 °C for one minute. Dissolution of CB[6] was observed, testifying to the rapid formation of the product. The major product was isolated by semi-preparative HPLC in 72% yield and identified as the expected asymmetric [3]semirotaxane 4, which was characterised by reverse-phase analytical HPLC, high-resolution

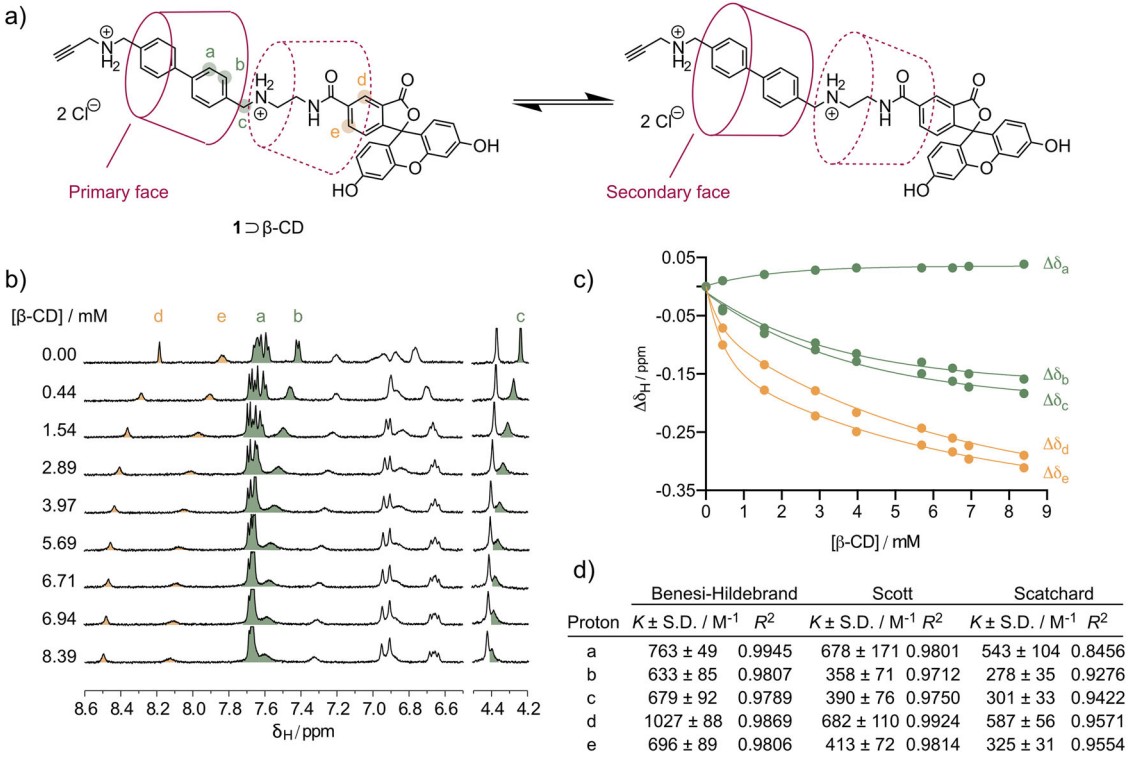

**Fig. 2 ¹H NMR titrations to determine the binding affinity and stoichiometry for the molecular inclusion 1⊃β-CD. a** Chemical structure of **1** with the position of β-CD in the **1**⊃β-CD inclusion complex. **b** Region of the ¹H NMR spectra of **1** when titrating with β-CD. **c** Chemical shift variations ($\Delta\delta_H = \delta_{free} - \delta_{complexed}$) of aromatic protons of **1**, as a function of β-CD concentration; [**1**] = 0.25 mM. **d** Calculated results for the binding constants interpreted by using the Benesi–Hildebrand[45], Scott[46], and Scatchard[47] models for a 1:1 equilibrium for which a linear relation involving $K$ and $\Delta\delta_H$ was found (expressed as the mean ($M^{-1}$) ± one standard deviation (S.D./$M^{-1}$) where $n = 5$ independent replicates).

electrospray ionisation mass spectrometry, and multinuclear NMR spectroscopy (Supporting Information Figs. S18–S26).

Following isolation of the [3]semirotaxane **4**, the full [3]rotaxane was synthesised by capping the axle through the formation of either the non-radioactive natGa- or the radioactive 68Ga-DFO complex to give [nat/68Ga]Ga-**4** (Supporting Information Figs. S27 and S28). Radiosynthesis using [68Ga][Ga(H₂O)₆]Cl₃ was accomplished by using standard radiolabelling methods to give [68Ga]Ga-**4** in a decay-corrected radiochemical yield (RCY) > 98% and with a radiochemical purity (RCP) of >95%, as measured by radio-analytical HPLC (radio-HPLC). Chromatographic measurements on [68Ga]Ga-**4** using radio-TLC developed by using a sodium citrate eluent (Fig. 3b, red trace) showed complete radiochemical conversion with all activity retained at the origin (retention factor, $R_f$ = 0.0-0.1), compared with the [68Ga]Ga-citrate control where all activity migrated to the solvent front ($R_f$ = 0.9–1.0). Analysis of [68Ga]Ga-**4** by radio-HPLC (Fig. 3c) gave a single peak that co-eluted with the electronic absorption peak (λ = 254 nm) observed for the authenticated sample of natGa-**4**.

Next, we expanded on this four-component strategy by testing a six-component reaction using one equivalent of each of the biphenyl dialkyne compound **2**, DFO-azido compound **3**, a fluorescein azido compound **5** (Supporting Information Scheme S3, Figs. S29–S35) and β-CD, combined with two equivalents of CB[6] to synthesise [4]semirotaxane **6** (Fig. 3a inset boxes, Supporting Information Scheme S4, Figs. S36–S39). The reaction was accomplished by heating the mixture of components in H₂O at 70 °C for one minute. In this reaction, two separate but likely sequential CB[6]-mediated click reactions occur on both sides of the biphenyl dialkyne **2**. Only products containing the β-CD macrocycle were observed, confirming the involvement of β-CD in accelerating the CB[6]-mediated triazole synthesis. This

reaction led to the synthesis of different [4]pseudorotaxane, [4]semirotaxane, and [4]rotaxane species in which three macro-cycles (one β-CD and two CB[6]) are trapped on an axle composed of a biphenyl guest molecule and two functional stoppers. In total, three products were identified. Two of the products featured identical stoppers at each end of the axle, giving the [4]pseudorotaxane with two DFO-chelates (reported previously)[26] and the [4]rotaxane with two fluorescein moieties (**7**). The identity of the symmetric bisfluorescein [4]rotaxane **7** was confirmed through independent synthesis and characterisa-tion of the compound (Supporting Information Scheme S5, Figs. S42–S52 and Table S1). The third product isolated was the desired asymmetric bifunctionalised [4]semirotaxane **6** with one DFO chelate and one fluorescein on each end of the axle (Fig. 3a). Compound **6** was purified by semi-preparative HPLC, isolated in 16% yield and fully characterised by analytical HPLC, high-resolution electrospray ionisation mass spectrometry (HRMS ESI), and multinuclear NMR spectroscopy.

Subsequent metal ion complexation reactions on [4]semi-rotaxane **6** with either nat Ga3+ or 68Ga3+ gave the corresponding [4]rotaxane natGa-**6** or [68Ga]Ga-**6** species in high isolated che-mical (>99%) and radiochemical yields (>98%) and with chemical or radiochemical purity >95% (Fig. 3, Supporting Information Figs. S40 and S41). Quantitative 68Ga-radiolabelling was accom-plished in <10 min at 23 °C. Characterisation of [68Ga]Ga-**6** was accomplished by using radio-chromatographic TLC and HPLC methods (Figs. 3b and 3d, respectively) which confirmed the chemical identity of the radiolabelled [4]rotaxane [68Ga]Ga-**6**. To the best of our knowledge, the CB[6]-mediated click strategy using six components has only been used to synthesise symme-trical rotaxanes, for which both stoppers at each end of the axle are identical[26,42,48]. Therefore, this reaction demonstrates the

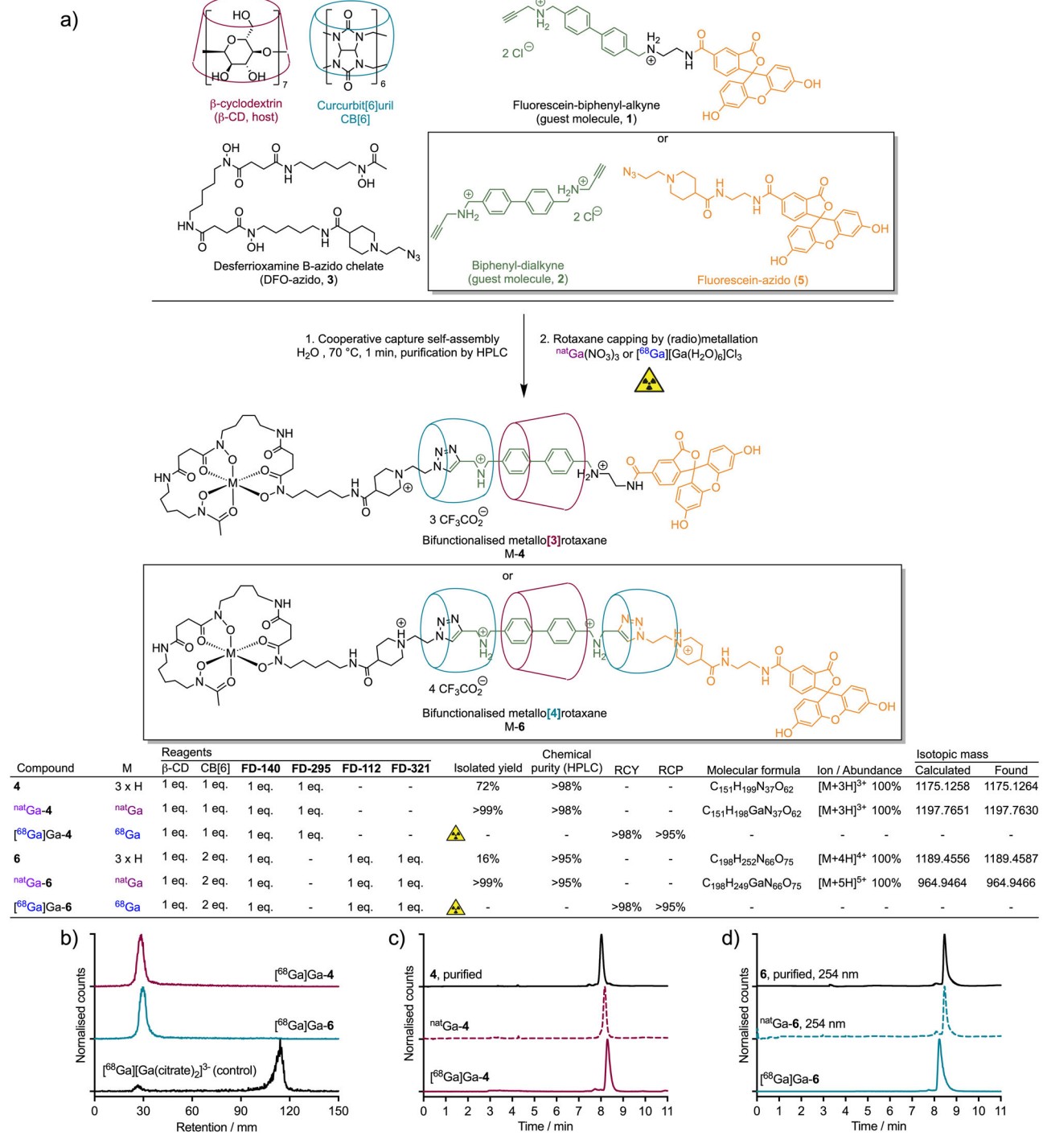

| Compound | M | Reagents | | | | | | Isolated yield | Chemical purity (HPLC) | RCY | RCP | Molecular formula | Ion / Abundance | Isotopic mass | |
|---|---|---|---|---|---|---|---|---|---|---|---|---|---|---|---|
| | | β-CD | CB[6] | FD-140 | FD-295 | FD-112 | FD-321 | | | | | | | Calculated | Found |
| **4** | 3 x H | 1 eq. | 1 eq. | 1 eq. | 1 eq. | - | - | 72% | >98% | - | - | $C_{151}H_{199}N_{37}O_{62}$ | $[M+3H]^{3+}$ 100% | 1175.1258 | 1175.1264 |
| nat Ga-**4** | natGa | 1 eq. | 1 eq. | 1 eq. | 1 eq. | - | - | >99% | >98% | - | - | $C_{151}H_{198}GaN_{37}O_{62}$ | $[M+3H]^{3+}$ 100% | 1197.7651 | 1197.7630 |
| [68Ga]Ga-**4** | 68Ga | 1 eq. | 1 eq. | 1 eq. | 1 eq. | - | - | ☢ - | - | >98% | >95% | - | - | - | - |
| **6** | 3 x H | 1 eq. | 2 eq. | 1 eq. | - | 1 eq. | 1 eq. | 16% | >95% | - | - | $C_{198}H_{252}N_{66}O_{75}$ | $[M+4H]^{4+}$ 100% | 1189.4556 | 1189.4587 |
| nat Ga-**6** | natGa | 1 eq. | 2 eq. | 1 eq. | - | 1 eq. | 1 eq. | >99% | >95% | - | - | $C_{198}H_{249}GaN_{66}O_{75}$ | $[M+5H]^{5+}$ 100% | 964.9464 | 964.9466 |
| [68Ga]Ga-**6** | 68Ga | 1 eq. | 2 eq. | 1 eq. | - | 1 eq. | 1 eq. | ☢ - | - | >98% | >95% | - | - | - | - |

**Fig. 3 The synthesis and characterisation of bifunctionalised asymmetric [3]rotaxane and [4]rotaxane constructs *via* cooperative capture. a** Chemical structures of the key reagents, the intermediate [3]semirotaxane **4** and [4]semirotaxane **6**, and the natGa- and 68Ga-radiolabelled metallo[3]rotaxane and metallo[4]rotaxane equivalents. **b** Radio-TLC chromatograms showing the successful radiosynthesis of [68Ga]Ga-**4** (maroon) and [68Ga]Ga-**6** (blue); Analytical reverse-phase C18 HPLC chromatograms showing the elution profile of the natGa3+ complexes (dashed lines) and the radiolabelled 68Ga3+ metallo[4]rotaxanes (solid lines) for (**c**) nat/68Ga-**4**, and (**d**) nat/68Ga-**6**. Note: chromatograms of the non-radiolabelled species were measured by electronic absorption at 254 nm.

feasibility of using a high number of different components to produce complex, asymmetric rotaxanes featuring multiple imaging components.

Both the four-component and six-component strategies for rotaxane synthesis were successful, but considering the different isolated yields (72% versus 16% yield, respectively), we decided to investigate the use of the four-component pathway to synthesise

biologically targeted, optically active rotaxane-based radiotracers (Fig. 4). As before, the CB[6]-β-CD-azide-alkyne click reaction was performed with **1**, a Glu-urea-Lys derivatised β-CD macro-cycle **8**, the DFO-azido **3** and CB[6], in $H_2O$ at 70 °C for one minute. The PSMA-targeted [3]semirotaxane **9** was isolated in 38% yield (Supporting Information Figs. S53–S61). Metal ion complexation of nat/68Ga by compound **9** gave the [3]rotaxane

**Fig. 4 Synthesis of the PSMA-targeted asymmetric [3]semirotaxane 9.** Conditions: **a** H₂O, 70 °C, 1 min, 38%.

species $^{nat}$Ga-**9** (Supporting Information Figs. S62 and S63) or [$^{68}$Ga]Ga-**9** in quantitative yield. Interestingly, in this case, one species was observed for the $^{nat}$Ga-**9** species, but under the radiolabelling conditions, three distinct isomers were identified (retention times for $^{nat}$Ga-**9**, $R_t = 8.28$ min, and [$^{68}$Ga]Ga-**9**, $R_t = 7.77$, 8.05 and 8.37 min, Supporting Information Fig. S64). The formation of multiple isomers for radiolabelled DFO compounds featuring other stereochemically active groups (like the Lys-urea-Glu biologically targeting vector, which has $S,S$ configuration) is a known phenomenon and is likely the result of diastereomer formation due to the various possible stereoisomers of the GaDFO complex[26,49]. In total, there are 16 potential geometric isomers formed when DFO coordinates to a metal ion to form an octahedral complex[49]. Differences between the nature of the starting reagents and in the reaction conditions between the $^{nat}$Ga and the $^{68}$Ga-radiolabelling reactions can also result in isomeric products. In addition, the fluorescein moiety is well-known to undergo structural changes forming several different isomers in aqueous conditions that are in equilibrium and depend heavily on the solution pH[50]. Finally, another type of stereoisomerism also arises in our [$^{nat/68}$Ga]Ga-**9** [4]rotaxane as a consequence of the mechanical bond. The presence of stereogenic centres in the oriented β-CD macrocycle (derived from enantiopure D-glucose, Fig. 5a) gives rise to two mechanically planar chiral diastereomers of the rotaxane, as illustrated in Fig. 5a[51,52]. As a result, all rotaxanes presented here, including [$^{nat/68}$Ga] Ga-**9**, were presumably formed as a mixture of the ($D$, $S_{mp}$) and ($D$, $R_{mp}$) mechanically planar epimers[53,54]. It is unclear what effect mechanical planar isomerisation (or molecular motion) might have on the physicochemical properties of the biologically targeted compounds, but attempts to isolate the different isomeric radioactive species were unsuccessful due to limitations arising from the very short half-life of the $^{68}$Ga radionuclide ($t_{1/2} = 67.71$ min).

Compound $^{nat}$Ga-**9** was further characterised by electronic absorption, electronic excitation, and fluorescence emission spectroscopy (Supporting Information Figs. S65 and S66, and Table S2). The maximum excitation and emission wavelengths of $^{nat}$Ga-**9** were determined to be 491.0 nm and 520.0 nm, respectively. The small Stokes shift (29 nm) is characteristic of fluorescein compounds, which undergo only a small change in geometry upon electronic excitation[14,55]. The fluorescence emission quantum yield of compound $^{nat}$Ga-**9** (Φ) was determined in 0.1 M NaOH relative to fluorescein (Φ = 0.93 in 0.1 M NaOH)[56] and found to be 0.70 ± 0.04 (Fig. 5b), which is within the range of previously reported quantum yields for fluorescein compounds[56,57].

Before performing cellular binding and uptake assays, the stability of compounds **4** and **9** and their $^{nat}$Ga-metallated equivalents $^{nat}$Ga-**4** and $^{nat}$Ga-**9** were studied in water (supporting information Figure S67). As anticipated, semirotaxanes **4** and **9** were not stable with half-lives of 24.61 h ± 2.00 h ($R^2 = 0.9863$) and 28.85 h ± 2.50 h ($R^2 = 0.9797$), respectively. Dethreading of the non-covalently bound CB[6] and β-CD macrocycles from the axle is the likely mechanism of semirotaxane degradation[26]. In contrast, the $^{nat}$Ga-complexed [3]rotaxanes, $^{nat}$Ga-**4** and $^{nat}$Ga-**9** were found to be stable, confirming that capping of the axle with a GaDFO stopper prevents dethreading of the macrocycles. Experimentally measured half-lives of $^{nat}$Ga-**4** and $^{nat}$Ga-**9** were 560.5 h ± 55.40 h ($R^2 = 0.9557$) and 254.7 h ± 19.50 h ($R^2 = 0.9685$), respectively. Given the short half-life of $^{68}$Ga the observed stability of the $^{nat}$Ga-**4** and $^{nat}$Ga-**9** rotaxanes was deemed acceptable for potential use in biological studies. Further studies demonstrated the stability of the [3]rotaxanes [$^{68}$Ga]Ga-**4** and [$^{68}$Ga]Ga-**9** in PBS and in human serum for 2 h at 37 °C (Supporting Information Tables S3 and S4) and that the $^{68}$Ga-radioactivity remained bound to the rotaxanes.

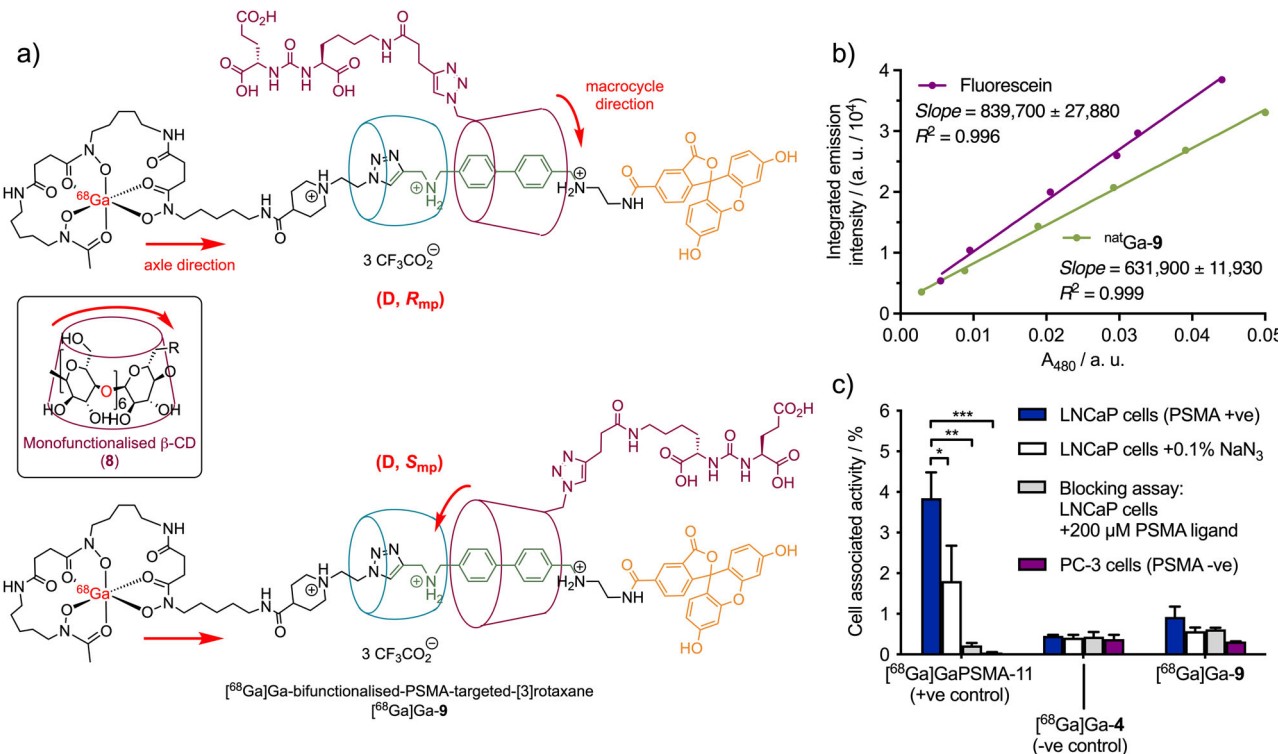

**Fig. 5 Structure, characterisation and biological evaluation of the [3]rotaxane [68Ga]Ga-9. a** Structure of the mechanically planar chiral stereoisomers for the [3]rotaxane [68Ga]Ga-**9** (the highest priority atoms in each component are highlighted in red). Note: Other components of the molecule, including the hexadentate GaDFO complex and the fluorescein group, can also exhibit geometric isomerism in solution. Stereochemical assignments were made in accordance with the Cahn-Ingold-Prelog rules and the methodology reported by Jamieson et al. where Ga has a higher priority than fluorescein, and clockwise rotation occurs for the primary face of the functionalised β-[D]-CD macrocycle[53,54]. Spectroscopic data associated with the [3]rotaxane natGa-**9** showing: **b** the integrated emission intensity (485–620 nm versus absorbance at the excitation wavelength (480 nm) for fluorescein (uses as a reference control) and natGa-**9**. **c** Cellular binding assay of [68Ga]Ga-PSMA-11 (used as a positive control), [68Ga]Ga-**4** (negative control), and [68Ga]Ga-**9** with both the LNCaP (PSMA +ve) and PC-3 (PSMA −ve) cell lines. Data were normalised to the total protein content and are presented as the percentage of activity bound to cells per 1 mg mL$^{-1}$ of protein. Error bars show one standard deviation (SD). Student's $t$-test: *) $P$-value > 0.05, **) $P$-value > 0.01, ***) $P$-value > 0.001, Where stated, cell media contained 0.1% azide to inhibit cellular internalisation, and protein content was measured by using a BCA assay. Blocking assays were conducted by using the LNCaP (PSMA +ve) cell line pre-treated with free PSMA ligand (200 µM) before the addition of radiotracers to saturate the available PSMA target protein.

Given the successful synthesis and stability studies, the specific binding of the PSMA-targeted bimodal [3]rotaxane [68Ga]Ga-**9** was investigated by using PSMA-expressing LNCaP prostate cancer cells. As a biological control, cellular binding assays were also performed by using the PSMA-negative PC-3 prostate cancer cells. The binding profile and percentage of cellular uptake of [68Ga]Ga-**9** were also compared against the established clinical-grade radiotracer [68Ga]Ga-PSMA-11 (positive control, Fig. 5c) and against the non-targeted [3]rotaxane [68Ga]Ga-**4** (used as a negative control)[26]. Uptake in LNCaP cells pretreated with 0.1% Na-azide was used to evaluate internalised *versus* surface-bound activity, and blocking studies using LNCaP cells pretreated with an excess of the free PSMA binding ligand (Supporting Information Fig. S68) were used to demonstrate specificity of binding to the PSMA target protein. First, cellular uptake of the positive control radiotracer [68Ga]Ga-PSMA-11 displayed the anticipated behaviour across all four cellular assays, confirming that the experimental methods were valid. High uptake of [68Ga]Ga-PSMA-11 was observed in LNCaP cells (Fig. 5c, blue bars), with a significant reduction when using Na-azide to inhibit cellular internalisation (Fig. 5c, white bars), and essentially no activity bound in either the blocking experiment or when using the PSMA negative PC-3 cells (Fig. 5c, grey and purple bars, respectively). For the PSMA-targeted [3]rotaxane [68Ga]Ga-**9**, generally lower cellular uptake was observed in the LNCaP model

compared with [68Ga]Ga-PSMA-11. However, higher uptake was observed for [68Ga]Ga-**9** in the LNCaP cells (0.92 ± 0.26%) compared with the non-targeted [3]rotaxane [68Ga]Ga-**4** (0.48 ± 0.02%, $P$-value = 0.085). These data provided an initial indication that specific uptake is observed when the Glu-urea-Lys is present in the rotaxane molecule. The targeted construct [68Ga]Ga-**9** also displayed the correct biological profile with respect to the Na-azide, PSMA-blocking, and PC-3 control assays, whereas for the non-targeted compound [68Ga]Ga-**4**, equivalent non-specific binding was observed across all experiments. For example, when 0.1% NaN₃ was added to the cell media, a 35% decrease in uptake was observed for [68Ga]Ga-**9** (Fig. 5c blue *versus* white bars, cell-associated activity of 0.57 ± 0.09% with NaN₃, $P$-value = 0.695). In the blocking study with the PSMA-free ligand, an approximate 30% decrease in comparison with the non-blocked sample was observed (Fig. 5c, [68Ga]Ga-**9**, blue *versus* grey bars, cell-associated activity of 0.62 ± 0.04% with the PSMA-free ligand, $P$-value = 0.175). Finally, when comparing the binding of [68Ga]Ga-**9** on LNCaP *versus* PC-3 cells, an approximately 3-fold decrease in uptake was observed (Fig. 5c, [68Ga]Ga-**9**, blue *versus* purple bars, the cell-associated activity of 0.32 ± 0.003% for PC-3 cells, $P$-value = 0.055). These results indicated that the cellular association of [68Ga]Ga-**9** was specific toward PSMA. However, [68Ga]Ga-**9** displayed a generally low level of cellular association that was

likely connected to a low affinity toward PSMA in comparison with the clinical grade radiotracer [68Ga]Ga-PSMA-11 (cell-associated activity $3.85 \pm 0.64\%$ in LNCaP cells, $P$-value = 0.008). Reduced cellular binding is to be expected for these first-generation supramolecular radiotracers when compared against the highly optimised and clinically approved [68Ga]Ga-PSMA-11 radiotracer (Locametz®, Novartis). Encouragingly, the cellular binding data illustrate that supramolecular dual-modality PET/OFI imaging agents featuring biologically active targeting vectors that display the correct biochemical behaviour can be produced by using the rotaxane architecture. Overall, these data support the further development of rotaxane-based imaging agents with improved binding profiles for future use in diagnostic imaging in vivo.

## Conclusions

Three new stoichiometrically precise asymmetrical rotaxane-based radiotracers were efficiently synthesised. The CB[6]-β-CD-azide-alkyne click reaction was shown to be a convenient approach to introduce functional capping groups on the rotaxane axle for the development of bimodal imaging agents featuring both a radioactive complex for PET and a fluorophore for OFI. This strategy was also successfully applied in the synthesis of the first PSMA-targeted bimodal rotaxane-based PET/OFI probe. In future developments, both the four-component or the six-component CB[6]-β-CD cooperative capture approach can be easily adapted to incorporate other molecules with additional features, such as alternative fluorophores for tuning the absorption and emission wavelengths, chelates for complexing different radioactive metal ions, antibody or peptide-based targeting vectors or chemotherapeutic drug molecules to create a diverse range of dual-modality, cancer-targeted agents for imaging and therapy. The possibility of attaching different functional groups at the end of the rotaxane axle will greatly expand the applicability of our rotaxane-based scaffold in drug/diagnostic probe design.

## Methods

Full details on the synthesis and characterisation of all compounds, including NMR spectra, high-resolution mass spectrometry data and HPLC chromatograms, as well as additional data from the radiochemistry and cellular experiments. are presented in Supplementary Note 1.

**Chemicals and solvents**. Unless otherwise stated, all other chemicals were of reagent grade and purchased from Sigma Aldrich (St. Louis, MO), Merck (Darmstadt, Germany), Tokyo Chemical Industry (Eschborn, Germany), abcr (Karlsruhe, Germany) or CheMatech (Dijon, France). Water (>18.2 MΩ·cm at 25 °C, Puranity TU 3 UV/UF, VWR International, Leuven, Belgium) was used without further purification. Solvents for reactions were of reagent grade and, where necessary, were dried over molecular sieves. Solvent evaporation was performed under reduced pressure by using a rotary evaporator (Rotavapor R-300, Büchi Labortechnik AG, Flawil, Switzerland).

**NMR spectroscopy**. $^1$H and $^{13}$C NMR spectra were measured in deuterated solvents on a Bruker AV-400 ($^1$H: 400 MHz, $^{13}$C: 100.6 MHz) or a Bruker AV-500 ($^1$H: 500 MHz, $^{13}$C: 125.8 MHz) spectrometer. Chemical shifts (δ) are expressed in parts per million (ppm) relative to the resonance of the residual solvent peaks, for example, with DMSO $\delta_H$ = 2.50 ppm and $\delta_C$ = 39.5 ppm with respect to tetramethylsilane (TMS, $\delta_H$ and $\delta_C$ = 0.00 ppm). Coupling constants ($J$) are reported in Hz. Peak multiplicities are abbreviated as follows: $s$ (singlet), $d$ (doublet), $dd$ (doublet of doublets), $t$ (triplet), $q$ (quartet), $m$ (multiplet), and $b$ (broadened). Two-dimensional NMR experiments, including $^1$H–$^1$H correlation spectroscopy (COSY), $^{13}$C–$^1$H heteronuclear single quantum coherence (HSQC) and $^{13}$C–$^1$H heteronuclear multiple bond correlation, and rotating frame Overhauser enhancement spectroscopy were performed to aid the assignment of the $^1$H and $^{13}$C spectra.

**Mass spectrometry**. HRMS (ESI) was measured by the mass spectrometry service at the Department of Chemistry, University of Zurich.

**Thin-layer chromatography**. Column chromatography was performed by using Merck silica gel 60 (63–200 μm) with eluents indicated in the experimental section. Standard TLC for synthesis employed Merck TLC plates silica gel 60 on an aluminium base with the indicated solvent system. The spots on TLC were visualised either by UV–Vis (254 nm) or by staining with KMnO$_4$.

**High-performance liquid chromatography**. Semi-preparative high-performance liquid chromatography (HPLC) was performed on a Rigol L-3000 system (Contrec AG, Switzerland) equipped with a UV–Vis (absorption measured at 220 nm and 254 nm), fitted with a reverse-phase VP 250/21 Nucleodur C18 HTec (21 mm ID × 250 mm, 5 μm) column. For all HPLC performed, solvent A = 18.2 MΩ cm H$_2$O + 0.1% TFA and solvent B = MeOH and the method used a flow rate of 7 mL min$^{-1}$ with a linear gradient of A: $t$ = 0 min 5% B; $t$ = 30 min 100% B; $t$ = 40 min 100% B.

Analytical HPLC experiments were performed by using Hitachi Chromaster Ultra Rs systems fitted with a reverse phase VP 250/4 Nucleodur C18 HTec (4 mm ID × 250 mm, 5 μm, Macherey–Nagel, Düren, Germany) column. This system was also fitted to a FlowStar$^2$ LB 514 radioactivity detector (Berthold Technologies, Zug, Switzerland) equipped with a 20 μL PET cell (MX-20-6, Berthold Technologies) for analysing radiochemical reactions. For all HPLC chromatograms shown, solvent A = 18.2 MΩ cm H$_2$O + 0.1% TFA and solvent B = MeOH and the method used a flow rate of 0.7 mL min$^{-1}$ with a linear gradient of A: $t$ = 0 min 5% B; $t$ = 1 min 5% B; $t$ = 10 min 100% B; $t$ = 11 min 100% B.

**Electronic absorption spectroscopy (UV–Vis)**. Electronic absorption spectra were recorded using a Nanodrop$^{TM}$ One$^C$ Microvolume UV–Vis Spectrophotometer (ThermoFisher Scientific, supplied by Witec AG, Sursee, Switzerland).

**Fluorescence emission spectroscopy**. Fluorescence measurements were carried out on a PerkinElmer LS50B fluorescence spectrometer using a 1 cm cell.

**Radioactivity**

*Caution*. Gallium-68 ($t_{1/2}$ = 67.71 min, $E_{max}(β^+)$=1899.1 keV, $I(β^+)$ = 87.72%, $Eγ$ = 1077.3 keV [$I$ = 3.2%], 511.0 keV [$I$ = 177.82%]) emits positrons and high-energy gamma rays. All operations must be performed by qualified personnel in an approved facility and following safety guidelines set forth by the local authorities and the Nuclear Regulatory Commission. Experimental manipulations should first be practised with non-radioactive samples, and researchers should follow the ALARA (As Low As Reasonable Achievable) protocols to minimise exposure to ionising radiation.

Gallium-68: [68Ga][Ga(H$_2$O)$_6$]Cl$_3$(aq.) was obtained from $^{68}$Ge/$^{68}$Ga-generators (Eckert&Ziegler, Model IGG100 Gallium-68 Generator), eluted with 0.1 M HCl(aq.). The eluted $^{68}$Ga activity was trapped and purified by using a strong cation exchange column (Strata-XC, [SCX], Eckert&Ziegler). [68Ga][Ga(H$_2$O)$_6$]Cl$_3$(aq.) was eluted from the SCX cartridge by using a solution containing 0.1 M HCl(aq.) and 5 M NaCl(aq.) (SCX eluent). The $^{68}$Ga stock solution was added as the limiting reagent to an aqueous reaction mixture buffered with NaOAc (0.2 M, pH4.4). Reactions were monitored by using instant thin-layer chromatography (radio-TLC). Glass-fibre iTLC plates impregnated with silica-gel (iTLC-SG, Agilent Technologies). Radio-TLC plates were developed in citrate buffer (1.0 M, pH4.5) and analysed on a radio-TLC detector (SCAN-RAM, LabLogic Systems Ltd, Sheffield, United Kingdom). Radiochemical conversion (RCC) was determined by integrating the chromatographic data obtained from the radio-TLC plate reader and determining both the percentage of the radiolabelled product (retained at the baseline with retention factor, $R_f$ = 0.0-1) and 'free' $^{68}$Ga (which elutes at the solvent front, $R_f$ = 0.9–1.0, as [68Ga][Ga(citrate)$_2$]$^{3-}$). Integration and data analysis were performed by using the software Laura version 5.0.4.29 (LabLogic). Small-molecule $^{68}$Ga-radiolabelled products were characterised by analytical radio-HPLC (method described above). $^{68}$Ga-labelled protein samples were characterised by manual SEC and automatic SEC–HPLC methods. Note that the UV–Vis detector and radioactivity detector were arranged serially with an offset time of approximately 0.10–30 min (depending on the ambient temperature). The identities of the radiolabelled compounds were confirmed by comparison of the retention times with an authenticated sample of the natural (non-radioactive) compounds that were used as standards.

**Quantification of radioactivity**. Fractions obtained from manual SEC and tissues collected from the animal experiments were measured on a gamma counter (HIDEX Automatic Gamma Counter, Hidex AMG, Turku, Finland) by using a counting time of 30 s and an energy window between 480–558 keV for $^{68}$Ga (511 keV emissions). Appropriate background and decay corrections were applied throughout. Activity measurements were performed by using a dose calibrator (ISOMED 2010 Activimeter, Nuklear-Medizintechnik Dresden GmbH, Germany).

**Cell culture**. All cells were cultured at 37 °C in a humidified 5% CO$_2$ atmosphere. Cell media was supplemented with foetal bovine serum (FBS, 10% ($v/v$), ThermoFisher Scientific) and penicillin/streptomycin (P/S, 1% ($v/v$) of penicillin

10,000 U/mL and streptomycin 10 mg mL$^{-1}$). LNCaP cells were pelleted (100 g, 10 min) and resuspended in media after trypsinisation.

*LNCaP cells.* The human prostate cancer cell line LNCaP was obtained from the American Type Culture Collection (ATCC-CRL-1740), Manassas, VA). Cells were cultured in RPMI-1640 (without phenol-red). Cells were grown by serial passage and were harvested by using trypsin (0.1%).

*PC-3 cells.* The human prostate cancer cell line PC-3 was obtained from the American Type Culture Collection (ATCC-CRL-1435), Manassas, VA). Cells were cultured in RPMI-1640 (without phenol-red). Cells were grown by serial passage and were harvested by using trypsin (0.5%).

**Cell binding assays**. Cells were harvested and distributed in Eppendorf tubes ($2.5 \times 10^6$ cells/vial) in media (270 µL) or media with sodium azide (270 µL, 0.1%). Radiolabelled compounds were prepared as described in the synthesis sections below, and aliquots of the purified compounds were diluted in cell media to obtain working stock solutions with an activity concentration of 100 kBq in 30 µL. To each Eppendorf tube was added a 30 µL aliquot of the stock solution of the radiotracer (final volume = 300 µL). After mixing for 1.5 h at 37 °C, the samples were centrifuged (2000 rpm, 4 °C, 3 min), and the cell pellet was washed with ice-cold PBS ($2 \times 1$ mL), keeping the samples on ice between washes. The radioactivity associated with each sample (cell pellet) was quantified by using the gamma counter. Experiments were performed in triplicate. Aliquots of the radioactive stocks (30 µL, 100 kBq) were added into three additional Eppendorf tubes, which did not contain cells, and were used as a control to measure the total activity. For control (blocking) studies, cells were harvested and distributed in Eppendorf tubes ($2.5 \times 10^6$ cells/vial) in media that contained free PSMA ligand (200 µM).

**Protein concentrations (bicinchoninic acid assay)**. The working reaction buffer was freshly prepared by adding $Cu(SO_4)_2$ (4%) to Pierce$^{TM}$ BCA Protein Assay Reagent A (ThermoFisher Scientific) in a 1:50 ratio. Cell lysate (5 µL) was added to the working reaction buffer (50 µL) in triplicate in a 96-well plate. The plate was incubated for 30 min at 37 °C. The absorbance was measured on Hidex-Sense Plate Reader using the BCA Assay programme. Absorbance was compared to a standard solution curve (using bovine serum albumin protein). The protein concentrations of cell lysates were normalised by dilution in saline.

**Statistical analysis**. Where appropriate, data were analysed by the unpaired, two-tailed Student's *t*-test. Differences at the 95% confidence level (* *P*-value < 0.05) were considered statistically significant. Note: (**) *P*-value < 0.01; (***) *P*-value < 0.001.

**Reporting summary**. Further information on research design is available in the Nature Portfolio Reporting Summary linked to this article.

## Data availability

All relevant data are presented in the main article or the supporting information.

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

## Acknowledgements

J.P.H. is supported by the Swiss National Science Foundation (SNSF Professorship PP00P2_163683 and PP00P2_190093) and the University of Zurich (UZH) for financial support. F.dO. received a Swiss Government Excellence Scholarship (ESKAS-Nr: 2017.0043). We thank all members of the Holland group for helpful discussions and continuous support. We thank Prof. Chenfeng Ke, Prof. Michal Juricek and Prof. Oliver Zerbe for their helpful discussions.

## Author contributions

F.dO. and J.P.H. designed all experiments, analysed the data, and wrote the paper. F.dO. conducted all experiments, including the synthesis, characterisation, radiochemistry, and cellular studies. J.P.H. assisted with radiochemistry and supervised the project.

## Competing interests

The authors declare no competing interests.
