## [Peer Review File · Communications Chemistry]

Asymmetric rotaxanes as dual-modality supramolecular imaging agents for targeting cancer biomarkersReviewers' comments:

Reviewer #1 (Remarks to the Author):

This paper submitted by Holland et al. describes the preparation of a series of rotaxanes ([3]- and [4]rotaxanes) with non-symmetric threads, employing cucurbit[6]uril and β -cyclodextrin as macrocycles, following a strategy already established by them (Angew. Chemie - Int. Ed. 61, e202204072 (2022)). The CB[6]- β -CD-azide-alkyne click triazole-formation was a convenient approach to introduce different functional capping groups on the thread, allowing the synthesis of bimodal imaging agents, having a radioactive complex for PET and a fluorophore for OFI. Titration experiments were performed for the optimization of the pseudorotaxanes assembly, calculating the association constants between the β -cyclodextrin macrocycle and a thread precursor. The synthesis and characterization of these novel derivatives are well conducted, counting that the purification of the final compounds should not be easy. This is a highly interesting paper, regarding the results reported. In my opinion, should be published once the authors have considered the corrections suggested hereafter:

1) In the case of the metallated rotaxane 9, the authors claimed the isolation of different isomers, which depends on various issues, including the formation of mechanoisomers due to the oriented β -CD macrocycle. In the case of rotaxanes M-4 and M-6, is it also possible to obtain the two orientational mechanoisomers (see: Org. Lett. 2004, 6, 26, 4869-4872; J. Am. Chem. Soc. 2005, 127, 35, 12186-12187). If during the titration experiments between the β -CD and compound 1, two conformations of the 1 \rightarrow β -CD inclusion complex were observed, differing the β -CD face orientation, the formation of the possible mechanoisomers of 4 and 6 is also feasible. Thus, are M-4 and M-6 pure compounds or mixture of the two possible epimers? The authors should clearly explain this point.

2) This reviewer has the curiosity if the author tried the formation of the corresponding [2]rotaxanes formed only with one CD macrocycle in the case of 4 and 6 analogs, or with the PSMA- β -CD macrocycle in the case of 9 (no addition of CB during the click reaction) (maybe these analogs are even formed during the reported protocol in minor amounts). The compounds formed (in the case of the analog of 9 without CBs) should potentially have the same activity against prostate cancer cells as the reported M-9. If not, which is the advantage (regardless the benefits on the synthesis due the cooperative effect between the CD and CB on the click reaction) of having CB macrocycles on the systems???

3) The authors measured the stability of rotaxanes Ga-4 and Ga-9 in water (560.5 h \pm 55.40 h for Ga-4; and 254.7 h \pm 19.50 h, for Ga-9). Did the authors think about this difference (2 fold) in the stability between the two systems? Moreover, the authors did not measure the stability of Ga-6. This experiment can be interesting in order to observe if the presence of 3 macrocycles in Ga-6 can increase the stability of the systems against degradation by protection of the macrocycles. One of the chemical consequences of the mechanical bond is the well-known protecting effect of the macrocycle over the functionalities present at the thread, reducing their reactivity and increasing their stability.

Minor point at the Supporting Information:

1) In Figure S9. Change the word "complex" for "compound" to name 1

2) In Figure S16. No figure S16C is commented in the legend.

3) Page 27: compound 14 should be referred as the corresponding TFA salt, not as a neutral amine.

4) Page 26 and Page 37. The authors should include the HPLC chromatograms for [⁶⁸Ga]Ga-4 and [⁶⁸Ga]Ga-6.

5) NMR data for compound Ga-9 is missed. The authors should include the spectra and data.

6) The HPLC chromatogram of semirotaxane 9 has a big impurity peak overlapping with the one of the compound. The authors claimed that the purity of this compound is >95%, same as [4]rotaxane 7, with a clear HPLC chromatogram. I suggest that the authors look on chromatogram of semirotaxane 9 to better calculate the purity (should it be less than 95%?)..

Reviewer #2 (Remarks to the Author):

The article by d'Orchymont and Holland describes the synthesis, radiolabeling and characterization of dual-labeled rotaxane compounds that combine optical fluorescence and PET imaging targeting the prostate-specific membrane antigen, a protein overexpressed by prostate cancer cells making it an attractive molecular target for prostate cancer diagnosis and therapy. This is an highly intriguing approach using supramolecular structures for the design of targeted imaging probes, which is of high interest to the readership of Chem Comm. The article is well-written and appropriate in length. The description of experiments is very detailed allowing their straightforward reproduction. The conclusions are in agreement with the presented results.

This reviewer was asked to comment on the radiochemistry part of the work. The authors use the well-known bifunctional DFO, which binds ^{68}Ga with high efficiency at room temperature. Labeling conditions follow standard protocols. Results of the stability studies are consistent with literature reports. DFO binds trivalent Gallium tightly so that corresponding radiocomplexes, as expected, do not show any decomposition in the stability measurements. Obviously, conjugation of DFO to supramolecular structures does not impact its complexation properties. Two minor corrections should be included:

1) Please include information about the molar activity of the radiolabeled compounds.

2) SI: In the description of the labeling experiments, the information about the buffer used is buried in the general section. In the description of the labelings, only the amount of activity and precursor is given. One has to guess that the remaining volume to end up with a total labeling volume of 250 μL has to be filled with buffer. An experienced researcher might be able to come to this conclusion but it might be better to rephrase this part to make it more clear to the reader.

Cellular uptake studies were also performed with a ^{68}Ga -labeled rotaxane construct to demonstrate PSMA-specific internalization. The experimental setup is excellent including a chemical as well as a biological control. The lower uptake compared to the clinically used PSMA-11 is not surprising considering the possible steric hindrance due to the supramolecular construct. From the results, it seems that the cell-associated activity is statistically significant compared to the controls. Adding this information to the manuscript would further improve the quality of the work. Pg 14, last sentence of the cell studies: "free PSMA ligand": Please include information which compound was used, PSMA-11? Or the rotaxane compound? Overall, the PSMA-specific uptake is promising indicating that such compounds may find radiopharmaceutical applications in the (near) future.

- Minor correction in "author contributions": Please remove the term "animal experiments".

- Please check the references thoroughly. For example, in some title chemical expression should be superscript. Or in some cases, authors are not correctly listed.

Reviewer #3 (Remarks to the Author):

This reviewer was asked to comment in particular on the imaging and radiopharmaceutical part of the manuscript.

The topic has a high clinical relevance. The combination of radioactive labels with fluorescent dyes for optical imaging is of increasing importance in the field of the surgical removal of pathological tissue. The presented manuscript deals with rotaxane based imaging agents in this context. Two challenging synthetic routes are presented to prepare new asymmetric ligands. This synthetic route was used to develop PSMA binding molecules that are suitable candidates for such an application. Preclinical characterization was performed, specifically to demonstrate that molecules generated on this asymmetric rotaxane-based platform are suitable for dual modality application in the field of prostate cancer.

These molecules are challenging regarding their synthesis, and the design requires an intense focus on

metabolic as well as pharmacokinetic properties. In this manuscript, the authors present new chemical approaches to address this issue and to design improved dual-labeled compounds with the aim to better control metabolic and PK properties.

After reviewing the overall concept, it can be said that this is an innovative novel platform that enables the transfer of the building blocks to other binding molecules, providing the perspective for future expansion of the technology. Hence, in this manuscript, the focus is less on specific radiopharmaceuticals than on the overall technology. The presented radiopharmaceutical binding PSMA has rather exemplary character.

A weakness is the specific structure of the PSMA binding moiety Glu-urea-Lsy, not taking into account that aromatic structures next to the Lysine are necessary to obtain an optimized tumor uptake. However, since the platform is the major focus in this work, this optimization could also be the subject of a follow-up work and is not explicitly requested within this context.

From the radiopharmaceutical and preclinical point of view, the conclusion and methodological interpretation are reasonable. The conclusion is sound and fully supported by the data presented.

This reviewer has only minor points and minor queries and supports a publication of this work.

- there are some concerns regarding the stability particularly related to its use as a radiopharmaceutical. Stability studies in water showed high stability of the radiolabeled compound. However, the stability of the radiolabeled molecule in human serum revealed some concerns regarding subsequent in vivo use.

Therefore, the issue of stability should be better discussed or, better, methodologically investigated in more detail. For example, serum stability data after 72 hours would be useful to fully demonstrate that a radiopharmaceutical application will be feasible. Particularly as it is a platform, such information is important at this stage of development. In addition, stability data with full plasma should be obtained.

- Moreover, from the perspective of a clinical application, the exact position of each single molecule part could be of crucial importance. Since the macrocycle is described to be able to slide on the axis of the whole molecule, there might be different orientations. An appropriate discussion with regard to a subsequent clinical application should be provided.

- an in vivo proof-of concept by e.g. PET imaging would significantly strengthen this work and should be taken into consideration. In particular, concerns regarding the in vivo stability and non-specific organ uptake of this method could be eliminated. Furthermore, in vivo data are necessary to support a major aim of this work: to better control PK and metabolic properties of dual modality compounds.

Prof. Dr. Jason P. Holland
SNSF Professor
Phone +41 44 635 39 90
Fax +41 44 635 68 02
jason.holland@chem.uzh.ch
<http://www.chem.uzh.ch/research/holland.html>
www.hollandlab.org

Zurich, 11th April 2023

Article revision: (COMMSCHEM-22-0585)

Referee: 1

This paper submitted by Holland et al. describes the preparation of a series of rotaxanes ([3]- and [4]rotaxanes) with non-symmetric threads, employing cucurbit[6]uril and β -cyclodextrin as macrocycles, following a strategy already established by them (Angew. Chemie - Int. Ed. 61, e202204072 (2022)). The CB[6]- β -CD-azide-alkyne click triazole-formation was a convenient approach to introduce different functional capping groups on the thread, allowing the synthesis of bimodal imaging agents, having a radioactive complex for PET and a fluorophore for OFI. Titration experiments were performed for the optimization of the pseudorotaxanes assembly, calculating the association constants between the β -cyclodextrin macrocycle and a thread precursor. The synthesis and characterization of these novel derivatives are well conducted, counting that the purification of the final compounds should not be easy. This is a highly interesting paper, regarding the results reported. In my opinion, should be published once the authors have considered the corrections suggested hereafter:

We thank the referee for the positive appraisal of our work. Indeed, the referee is quite right that the purification of these compounds is quite tricky but can be achieved using preparative HPLC methods.

- 1) In the case of the metallated rotaxane 9, the authors claimed the isolation of different isomers, which depends on various issues, including the formation of mechanoisomers due to the oriented β -CD macrocycle. In the case of rotaxanes M-4 and M-6, is it also possible to obtain the two orientational mechanoisomers (see: Org. Lett. 2004, 6, 26, 4869–4872; J. Am. Chem. Soc. 2005, 127, 35, 12186–12187). If during the titration experiments between the β -CD and compound 1, two conformations of the 1 \supset β -CD inclusion complex were observed, differing the β -CD face orientation, the formation of the possible mechanoisomers of 4 and 6 is also feasible. Thus, are M-4 and M-6 pure compounds or mixture of the two possible epimers? The authors should clearly explain this point.

We thank the referee for providing the additional two references. We have included these two as citations in our revised manuscript. We note that the original submission already made note that mechanoisomers exist for all the rotaxanes that we report. Nevertheless, we have highlighted this point more clearly. The text on page 12 now reads:

Finally, another type of stereoisomerism also arises in our [^{nat/68}Ga]Ga-9 [4]rotaxane as a consequence of the mechanical bond. The presence of stereogenic centres in the oriented β -CD macrocycle (derived from enantiopure *D*-glucose, **Figure 3A**) gives rise to two mechanically

planar chiral diastereomers of the rotaxane, as illustrated in **Figure 3A**.^{51,52} As a result, **all rotaxanes presented here**, including [^{nat/68}Ga]Ga-9, were presumably formed as a equimolar mixture of the (*D*, *S*_{mp}) and (*D*, *R*_{mp}) mechanically planar epimers.^{53,54}

- 2) This reviewer has the curiosity if the author tried the formation of the corresponding [2]rotaxanes formed only with one CD macrocycle in the case of 4 and 6 analogs, or with the PSMA-β-CD macrocycle in the case of 9 (no addition of CB during the click reaction) (maybe these analogs are even formed during the reported protocol in minor amounts). The compounds formed (in the case of the analog of 9 without CBs) should potentially have the same activity against prostate cancer cells as the reported M-9. If not, which is the advantage (regardless the benefits on the synthesis due the cooperative effect between the CD and CB on the click reaction) of having CB macrocycles on the systems???

There is no click reaction without the CB[6] molecule. Beta-CD alone is incapable of forming a triazole and as such, the suggested [2]rotaxanes without CB[6] cannot be made via this chemical route. Mock demonstrated in 1983 that the CB[6] catalyses the triazole formation and this reaction is well-known. It was later used by Stoddart and co-workers (see Cooperative capture synthesis: yet another playground for copper-free click chemistry. *Chem. Soc. Rev.* **45**, 3766–3780 (2016)) to develop the rotaxane platform which we now elaborated for potential applications in molecular imaging. We also performed all the control reactions as part of a separate study but no rotaxane was observed without CB[6].

- 3) The authors measured the stability of rotaxanes Ga-4 and Ga-9 in water (560.5 h ± 55.40 h for Ga-4; and 254.7 h ± 19.50 h, for Ga-9). Did the authors think about this difference (2 fold) in the stability between the two systems? Moreover, the authors did not measure the stability of Ga-6. This experiment can be interesting in order to observe if the presence of 3 macrocycles in Ga-6 can increase the stability of the systems against degradation by protection of the macrocycles. One of the chemical consequences of the mechanical bond is the well-known protecting effect of the macrocycle over the functionalities present at the thread, reducing their reactivity and increasing their stability.

The referee makes an interesting observation about the potential difference in stability of the Ga-4 and Ga-9 complexes. We did not investigate this further since both compounds are sufficiently stable on the time scale of our radiosynthesis or on potential further biological studies with the corresponding ⁶⁸Ga-complexes where experiments should be completed within about 1 – 2 half-lives.

Since the synthesis method to produce Ga-6 was found to be more cumbersome than the simplified route that gave Ga-4 or Ga-9, we decided to pursue the biological studies using the stoichiometrically reduced 1:1:1:1 system only. Therefore, stability studies on Ga-6 are not required and at this point, we cannot resynthesize the molecule to obtain these data because Fasutine has now left to begin her postdoc. We do note that in our previous work, we did perform long term stability measurements on a related compound featuring a symmetric axle

with two GaDFO complexes (ACIE e202204072) where no degradation was observed over 96 h in solution (H₂O) and over 3 years as a lyophilised powder. These data are reproduced here for simplicity but full details are available online.

Stability studies on [4]pseudorotaxane **3** and metallo[4]rotaxanes ^{nat}Ga-**3** and ^{nat}Zr-**3**⁺.

Fresh solutions of **3**, ^{nat}Ga-**3** and ^{nat}Zr-**3**⁺ were prepared in H₂O and left at 23 °C for 96 h.

Figure S44. HPLC traces recorded at 254 nm at various time points for (a) **3**, (b) ^{nat}Ga-**3**, and (c) ^{nat}Zr-**3**⁺.

This figure supports main figure 2e

Figure S45. HPLC traces recorded at 254 nm after being stored as a lyophilised powder at 23 °C for 3 years.

We appreciate that these data are not a precise indicator of the stability of our new Ga-6 asymmetric rotaxane, but given that the compound is not pertinent to the study, we ask for the referees understanding in this matter.

Regarding the statement, ‘One of the chemical consequences of the mechanical bond is the well-known protecting effect of the macrocycle over the functionalities present at the thread, reducing their reactivity and increasing their stability. Simply, this has not been fully tested and we would have reservations in applying this general statement to a broader set of rotaxanes with different chemical functionalities. Future studies from our group will report on this matter but the topic is beyond the scope of the current manuscript.

Minor point at the Supporting Information:

- 1) In Figure S9. Change the word “complex” for “compound” to name 1

Done

- 2) In Figure S16. No figure S16C is commented in the legend.

Thank you very much for spotting this. We have added S16C into the legend.

3) Page 27: compound 14 should be referred as the corresponding TFA salt, not as a neutral amine.

Done. We have added the TFA to the figures.

4) Pag 26 and Pag 37. The authors should include the HPLC chromatograms for [68Ga]Ga-4 and [68Ga]Ga-6.

These chromatograms were already presented in the original submission and appear in the Main Text, Figure 2. We have not duplicated them in the ESI.

5) NMR data for compound Ga-9 is missed. The authors should include the spectra and data.

We do not have NMR data for the natGa complexes. The spectra broaden dramatically because of the presence of the metal ion, and the formation of potentially up to 16 different stereoisomers in the GaDFO complex.

6) The HPLC chromatogram of semirotaxane 9 has a big impurity peak overlapping with the one of the compound. The authors claimed that the purity of this compound is >95%, same as [4]rotaxane 7, with a clear HPLC chromatogram. I suggest that the authors look on chromatogram of semirotaxane 9 to better calculate the purity (should it be less than 95%?).

We thank the referee for spotting this. It was likely a cut/paste error. We have reintegrated the HPLC chromatogram in Figure S53 and to the best of our abilities estimate a purity of ~90%. We note that it is very hard to obtain an accurate number here because the two peaks overlap significantly. Efforts to separate these out with different chromatographic conditions were unsuccessful.

Referee: 2

The article by d'Orchymont and Holland describes the synthesis, radiolabeling and characterization of dual-labeled rotaxane compounds that combine optical fluorescence and PET imaging targeting the prostate-specific membrane antigen, a protein overexpressed by prostate cancer cells making it an attractive molecular target for prostate cancer diagnosis and therapy. This is an highly intriguing approach using supramolecular structures for the design of targeted imaging probes, which is of high interest to the readership of Chem Comm. The article is well-written und appropriate in length. The description of experiments is very detailed allowing their straightforward reproduction. The conclusions are in agreement with the presented results.

We thank the referee for the positive appraisal of our work.

This reviewer was asked to comment on the radiochemistry part of the work. The authors use the well-known bifunctional DFO, which binds 68Ga with high efficiency at room temperature. Labeling conditions follow standard protocols. Results of the stability studies are consistent

with literature reports. DFO binds trivalent Gallium tightly so that corresponding radiocomplexes, as expected, do not show any decomposition in the stability measurements. Obviously, conjugation of DFO to supramolecular structures does not impact its complexation properties. Two minor corrections should be included:

1) Please include information about the molar activity of the radiolabeled compounds.

Done. We have added these data to the ESI. The inserted text reads:

(ESI Page 26) After optimisation [⁶⁸Ga]Ga-4 was isolated with a molar activity of 20 MBq nmol⁻¹ (measured with titration experiments, $R^2 = 0.8648$).

(ESI Page 52) After optimisation [⁶⁸Ga]Ga-9 was isolated with a molar activity of 17 MBq nmol⁻¹ (measured with titration experiments, $R^2 = 0.9892$).

2) SI: In the description of the labeling experiments, the information about the buffer used is buried in the general section. In the description of the labelings, only the amount of activity and precursor is given. One has to guess that the remaining volume to end up with a total labeling volume of 250 μ L has to be filled with buffer. An experienced researcher might be able to come to this conclusion but it might be better to rephrase this part to make it more clear to the reader.

The referee is correct. The data could be calculated from the information that we already provided but we have now edited the section to read as follows (please note that equivalent changes were made in the other radiosynthesis sections for 68Ga-6 and 68Ga-9).

Radiolabelling reactions to prepare [⁶⁸Ga]Ga-4 were accomplished by the addition of an aliquot of [⁶⁸Ga][Ga(H₂O)₆]Cl₃(aq.) stock solution (~12 MBq, diluted in H₂O to ~190 mL) to an aqueous solution of 4 (10 μ L of 1 mM stock in H₂O) buffered with NaOAc (0.2 M, pH4.4, 50 mL) with a total reaction volume of 250 μ L.

Cellular uptake studies were also performed with a 68Ga-labeled rotaxane construct to demonstrate PSMA-specific internalization. The experimental setup is excellent including a chemical as well as a biological control. The lower uptake compared to the clinically used PSMA-11 is not surprising considering the possible steric hindrance due to the supramolecular construct. From the results, it seems that the cell-associated activity is statistically significant compared to the controls. Adding this information to the manuscript would further improve the quality of the work. Pg 14, last sentence of the cell studies: “free PSMA ligand”: Please include information which compound was used, PSMA-11? Or the rotaxane compound? Overall, the PSMA-specific uptake is promising indicating that such compounds may find radiopharmaceutical applications in the (near) future.

We thank the referee for noting our promising cellular data. Overall, the statistically analysis of the cellular uptake profiles for 68Ga-4 and 68Ga-9 showed that there is a difference when

compared with the ⁶⁸Ga-PSMA-11 positive control. However, visualising these differences on the plot in Figure 3C is not so simple. Therefore, we decided to add the *P*-values into the text.

We have now included a picture of the chemical structure of the small-molecule PSMA binding ligand used in blocking experiments in cells in supporting information **Figure S68**.

- Minor correction in “author contributions”: Please remove the term “animal experiments”.

Done. Thanks for spotting this.

- Please check the references thoroughly. For example, in some title chemical expression should be superscript. Or in some cases, authors are not correctly listed.

Done.

Referee: 3

This reviewer was asked to comment in particular on the imaging and radiopharmaceutical part of the manuscript.

The topic has a high clinical relevance. The combination of radioactive labels with fluorescent dyes for optical imaging is of increasing importance in the field of the surgical removal of pathological tissue. The presented manuscript deals with rotaxane based imaging agents in this context. Two challenging synthetic routes are presented to prepare new asymmetric ligands. This synthetic route was used to develop PSMA binding molecules that are suitable candidates for such an application. Preclinical characterization was performed, specifically to demonstrate that molecules generated on this asymmetric rotaxane-based platform are suitable for dual modality application in the field of prostate cancer.

These molecules are challenging regarding their synthesis, and the design requires an intense focus on metabolic as well as pharmacokinetic properties. In this manuscript, the authors present new chemical approaches to address this issue and to design improved dual-labeled compounds with the aim to better control metabolic and PK properties.

After reviewing the overall concept, it can be said that this is an innovative novel platform that enables the transfer of the building blocks to other binding molecules, providing the perspective for future expansion of the technology. Hence, in this manuscript, the focus is less on specific radiopharmaceuticals than on the overall technology. The presented radiopharmaceutical binding PSMA has rather exemplary character.

We thank the reviewer for their work and for the positive appraisal of our science.

A weakness is the specific structure of the PSMA binding moiety Glu-urea-Lys, not taking into account that aromatic structures next to the Lysine are necessary to obtain an optimized tumor

uptake. However, since the platform is the major focus in this work, this optimization could also be the subject of a follow-up work and is not explicitly requested within this context.

The referee is absolutely right that optimal binding to the PSMA pocket requires refining intermolecular interactions with aromatic residues, which are not present in our current structures. We note that the current work focuses on synthetic optimisation and not radiotracer performance. This approach will be taken in future studies, drawing from structures like PSMA-617.

From the radiopharmaceutical and preclinical point of view, the conclusion and methodological interpretation are reasonable. The conclusion is sound and fully supported by the data presented.

Thanks

This reviewer has only minor points and minor queries and supports a publication of this work.

- there are some concerns regarding the stability particularly related to its use as a radiopharmaceutical. Stability studies in water showed high stability of the radiolabeled compound. However, the stability of the radiolabeled molecule in human serum revealed some concerns regarding subsequent in vivo use.

Therefore, the issue of stability should be better discussed or, better, methodologically investigated in more detail. For example, serum stability data after 72 hours would be useful to fully demonstrate that a radiopharmaceutical application will be feasible. Particularly as it is a platform, such information is important at this stage of development. In addition, stability data with full plasma should be obtained.

We understand the reviewers point but we note that the stability of ^{68}Ga -4 and ^{68}Ga -9 decreased to 92 and 94% after 2 h. In the context of ^{68}Ga radiopharmaceuticals this is not concerning change. We also note that the 72 h time point is not relevant or feasible with ^{68}Ga .

- Moreover, from the perspective of a clinical application, the exact position of each single molecule part could be of crucial importance. Since the macrocycle is described to be able to slide on the axis of the whole molecule, there might be different orientations. An appropriate discussion with regard to a subsequent clinical application should be provided.

We agree with the referee that having a single isolated isomer is potentially important for clinical applications but these compounds are nowhere near the clinic. Mechanical motion might also be important but the effect that this would have in vivo is unclear. We mentioned this in the following sentence:

(Page 12): “It is unclear what effect mechanical planar isomerisation (or molecular motion) might have on the physicochemical properties of the biologically targeted compounds”.

- an *in vivo* proof-of concept by e.g. PET imaging would significantly strengthen this work and should be taken into consideration. In particular, concerns regarding the *in vivo* stability and non-specific organ uptake of this method could be eliminated. Furthermore, *in vivo* data are necessary to support a major aim of this work: to better control PK and metabolic properties of dual modality compounds.

The PET experiments *in vivo* are unjustified on ethical grounds. The cellular data provide sufficient indication that the tracer will not bind as well as the clinical radiopharmaceutical Ga-PSMA-11 and therefore animal studies were not performed. Further chemical optimisation (along the lines described by this referee regarding aromatic residue interactions) are needed first before *in vivo* experiments should be considered. If we can make it that far, then absolutely, PK profiling, PET/optical co-registration, and metabolite analysis would be incorporated into the study design.

REVIEWERS' COMMENTS:

Reviewer #1 (Remarks to the Author):

The authors response all of the request changes and suggestions. In my opinion the manuscript can be accepted without further changes.

Reviewer #3 (Remarks to the Author):

All comments have been sufficiently addressed and explained. The manuscript might now be published in its revised form.